# Prospective clinical performance of CoVarScan in identifying SARS-CoV-2 Omicron subvariants

Kenneth Zhu,[1] Manoj Sah,[1] Lenin Mahimainathan,[1] Yan Liu,[1] Chao Xing,[1] Karen Roush,[2] Andrew Clark,[1] Jeffrey SoRelle[1]

**ABSTRACT**    The purpose of this work was to evaluate the performance of CoVarScan, a multiplex fragment analysis approach, in identifying severe acute respiratory syndrome coronavirus 2 (SARS-CoV-2) variants of the Omicron lineage rapidly and accurately. The ability to identify variants with high fidelity and low turnaround time is important both epidemiologically and clinically for pandemic monitoring and therapeutic monoclonal antibody (mAb) selection. Currently, the gold-standard test for this task is whole-genome sequencing (WGS), which is prohibitively expensive and/or inaccessible due to equipment requirements for many laboratories. Omicron variants have been closely related, so the ability of genotyping tests to differentiate them is an important, outstanding question. CoVarScan uses PCR targeting eight SARS-CoV-2 mutational hot spots. In total, 4,918 SARS-CoV-2-positive cases between 17 December 2021 and 31 January 2024 were included in the analysis. CoVarScan achieved 96.5% concordance with WGS and could detect unique mutational signatures for BA.1, BA.2, BA.2.12.1, BA.4/ BA.5, BA.2.75, XBB, and BA.2.86. These are the major variants of concern (VOCs) that have dominated since Omicron originally appeared in December 2021. Lastly, based on panel design, we predict a unique mutational pattern for the newly emergent, highly mutated variant BA.2.87. CoVarScan can rapidly, accurately, and cost-effectively identify all Omicron variants in a scalable manner. Furthermore, CoVarScan does not require design alterations to detect new VOCs. CoVarScan performs as accurately as WGS with higher sensitivity, allowing its use as a tool to quickly identify variants for epidemiological surveillance and clinical decision-making in the selection of effective therapeutic mAbs.

**IMPORTANCE**    Almost 5 years since the start of the pandemic, severe acute respiratory syndrome coronavirus 2 (SARS-CoV-2) variants of concern continue to emerge, with mutations conferring new properties like increased transmissibility and resistance to therapeutic monoclonal antibodies and vaccines. Conventionally, whole-genome sequencing (WGS) has characterized new SARS-CoV-2 variants, but results come too late for clinical actionability. WGS suffers from high failure rates for samples with low viral RNA and is inaccessible for lower-resource laboratories. As new variants like Omicron appear, it is necessary to develop rapid and accurate testing to distinguish between variants. Fast and accurate identification of sensitive viral lineages would allow tailored use of monoclonal antibodies that may otherwise have been pulled from the market due to rising overall resistance. Rapid results also allow public health officials to make policy decisions in time to reduce morbidity and mortality for sensitive populations such as patients who are immunocompromised or have significant medical comorbidities.

**KEYWORDS**    COVID-19, SARS-CoV-2, multiplex PCR, capillary electrophoresis, fragment analysis, Omicron

Address correspondence to Jeffrey SoRelle, Jeffrey.SoRelle@UTSouthwestern.edu.

J.S. is entitled to royalties from a pending patent application on technology described in this paper.

See the funding table on p. 9.

Severe acute respiratory syndrome coronavirus 2 (SARS-CoV-2) variants of concern (VOCs) continue to emerge due to evolutionary pressures. While SARS-CoV-2 mutates at a relatively slower rate (one mutation per genome every 2 weeks) (1), the Omicron surge arose from a significantly different variant, and there have been gradual and large genomics shifts since.

VOC-associated changes to the receptor-binding domain (RBD) cause increased resistance to antibodies (from previous infection or vaccination), reduced efficacy of treatments/tests, or predicted increase in transmissibility/severity. Most new mutations have conferred increased transmissibility, increased resistance to therapeutic monoclonal antibodies (mAbs), and altered symptom profiles. The original Omicron variant (BA.1) caused pharyngitis, and more recently, the XBB.1.16 variant was found to cause conjunctivitis (2, 3). Testing must rapidly and accurately identify specific variants to best inform clinical and epidemiological responses (4, 5). Conventionally, whole-genome sequencing (WGS) has characterized new SARS-CoV-2 variants, but results come too late to direct clinical management. WGS also yields lower depth of coverage for samples with low viral RNA (Ct value > 30) (6). There are two emergency use authorization (EUA)-approved variant tests currently, and both use sequencing technology from ClearDx and LabCorps. The limit of detection of the ClearDx and LabCorps tests are 28–30 and 25, respectively. Furthermore, WGS is often inaccessible due to limited equipment, financial resources, and specialized bioinformatics expertise (5–9).

With so many mutations in BA.1, genotyping and next-generation sequencing (NGS) tests had difficulty distinguishing subsequent variants, and the conventionally used antibody cocktails were ineffective (10). Sotrovimab was effective against Omicron BA.1 by reducing hospitalization (11). This mAb targets a highly conserved spike epitope rather than the rapidly mutating RBD of Spike and effectively neutralizes BA.1 (11). However, after just a short few months, a February 2022 study showed BA.2 to be genetically distant enough to have robust resistance to sotrovimab (12). As of 5 April 2022, sotrovimab was pulled from the market by the US Food and Drug Administration (FDA) to treat coronavirus disease 2019 (COVID-19) in any part of the United States due to resistance by BA.2 (https://www.fda.gov/drugs/drug-safety-and-availability/fda-updates-sotrovimab-emergency-use-authorization). The problem is this causes shortages of the only other available antibody, bebtelovimab, when as much as 90%+ of the population could have benefited. For any neutralizing mAb, the FDA withdraws EUA when the frequency of a neutralization-resistant variant rises above 10%.

While it is judicious to discontinue the use of ineffective treatments, proportions of SARS-CoV-2 strains are not uniform across the country and access to mAbs may be denied to patients who can benefit from treatment. This fact illustrates the importance of accurate and rapid variant identification for appropriate clinical intervention. To provide the benefit of longer access to appropriate therapeutics, rapid variant results are needed. Additionally, rapid genotyping would be much cheaper and faster than sequencing but would still identify significant divergence from previous variants, the principal concern for public health decision-making (7).

Our previous work on an eight-plex genotyping COVID-19 variant test (CoVarScan) by fragment analysis could provide such a benefit (13). Therefore, this study was performed to evaluate the accuracy of CoVarScan against Omicron subvariants from 2022 to 2024, including BA.2, BA.2.12.1, BA.4/BA.5, XBB, HV.1, and BA.2.86. We propose that CoVarScan provides sufficient resolution for clinically actionability and faster, cheaper, more sensitive epidemiology compared with WGS.

## MATERIALS AND METHODS

### Clinical specimens

Assay testing used SARS-CoV-2-positive nasopharyngeal or nasal specimens from the University of Texas Southwestern Medical Center pathology service line. Positivity was determined by one of four methods: RT-PCR using the Alinity *m* SARS-CoV-2 assay

(Abbott Molecular), Xpert Xpress SARS-CoV-2 assay or Xpert Xpress SARS-CoV-2 Flu A/B RSV assay (Cepheid), or isothermal amplification (IDNOW, Abbott Diagnostics). All specimens were placed and stored in viral transport media (Remel, Thermo Scientific) or Abbott Universal Collection Kit media. The timeframe of this study ranged from the date of clinical implementation to data analysis period 17 December 2021 to 31 January 2024 to study CoVarScan's performance in a real-world clinical setting with subvariants of Omicron, the dominant variant at the time. Our analysis used prospective SARS-CoV-2-positive specimens collected from asymptomatic, symptomatic, and confirmed COVID-19 patients. SARS-CoV-2 WGS was performed using the Swift Normalase Amplicon Panel as previously described (14).

### Inclusion criteria for validation

Specimens required successful WGS results for inclusion to ensure that a comparator method was available. As a quality control measure, specimens included in the analysis had RT-qPCR cycle threshold (CT) values less than 35 (Abbott Alinity M SARS-Cov-2 assay) or a positive qualitative result from either Cepheid or IDNOW COVID-19 assays. Specimens used are largely the same (in terms of patient population, nasopharyngeal or nasal source, quality, and method of determination of positivity) as those described in our previous study detailing the validation of CoVarScan (13).

### CoVarScan assay design

CoVarScan specifics are detailed in our previous study (13). In summary, eight amplicons of different sizes and colors were generated via RT-PCR using fluorophore-labeled primers. Amplicons were separated by capillary electrophoresis (ABI 3730XL, Applied Biosystems). Positive results were confirmed using WGS. To blind WGS results from the reviewers, sequencing was performed 1 week after examination and interpretation of results from fragment analysis. Furthermore, a separate group performed bioinformatic analyses blinded to CoVarScan results. As detailed previously, fragment analysis targeted defined hotspot regions of the spike RBD, recurrently deleted regions of the N-terminus, ORF1A, and ORF8 genes to sufficiently differentiate Omicron subvariants.

### Interpretive criteria

Samples were considered positive if capillary electrophoresis peaks were (1) the expected size (x-axis), (ii) 50 fluorescence units (double background fluorescence), and (iii) the HEX:FAM or FAM:HEX signal was 10:1 to call an allele as mutant or wild-type (WT). A minimum of three peaks were required for a result, and if only three were present and a variant could be distinguished from these targets, then a qualifier of likely was added before the variant identification. Full details can be found in the supplemental material of our previous study describing CoVarScan (13).

## RESULTS

### Resolving discordant results

A total of 4,918 specimens met the criteria for inclusion in validation studies. CoVarScan achieved 94.8% concordance with WGS upon initial review. However, a disproportionately high number ($n = 22$) of discordant results were BA.2, with most ($n = 20$) interpreted as BA.2.12.1 by WGS. Conversely, several CoVarScan BA.2.12.1 results ($n = 22$) were interpreted as BA.2.12 by WGS. To investigate these patterns, the original CoVarScan and raw WGS data corresponding to each case of discordance were re-examined with Nextclade (https://clades.nextstrain.org) using the most updated version of Pangolin to assign a variant classification. Many (27/44) originally discordant cases were resolved and found to be concordant through this process. We concluded the WGS results for these specimens were called incorrectly by the Pangolin lineage caller of the time.

There were 13 instances where a pattern was identical (BA.2.12.1 vs BA.2.75), and it required education and re-training to explain that BA.2.12.1 was not circulating when

BA.2.75 was present (Figure 1). Lastly, one clerical error (BA.2 CoVarScan vs BA.1 by WGS) was resolved upon review of raw data.

In some cases, final assigned results were incorrect (overcalled) by the Pangolin lineage software. In eight cases, specimens interpreted as BA.2 by CoVarScan were discordant with WGS (all BA.2.12.1). However, the key differentiating mutation, S:L452Q, produced no signal by CoVarScan. Coverage of L452Q mutation is required to distinguish between BA.2 and BA.2.12.1. Thus, these eight cases were reclassified as concordant since BA.2 is the parent lineage of BA.2.12.1. Lastly, two CoVarScan BA.2 cases were discordant with WGS (BA.2.12.1), but neither of the BA.2.12.1 lineage defining mutations (L452Q and S704L) were covered. Thus, these cases were excluded from analysis as it was not possible to distinguish between BA.2.12 and BA.2.12.1 (Figure 2). Interestingly, one specimen was interpreted as XBF via WGS and was interpreted as BA.2.75 via CoVarScan. This unique case was counted as discordant, despite the S gene sequences for both XBF and BA.2.75 being identical. The XBF is a rare recombinant between BA.5 and BA.2.75 sublineages, which most likely behaves like a BA.2.75 variant as the spike mutations come from BA.2.75 (15). Thus, detecting rare recombination events could be challenging if they have the same mutational profile as prevailing variants.

## Accuracy of CoVarScan

Table 1 organizes concordant (underlined) and discordant results by subvariants called by both CoVarScan and WGS after resolving for discrepancies due to technical and bioinformatic reasons. With 4,744 concordant and 174 discordant results, we found an aggregate accuracy of 96.5% (positive percent agreement) with a 3.5% discrepancy rate (negative percent agreement) across a total of 4,918 clinical respiratory samples tested by CoVarScan and WGS as the comparator method (Tables 2 and 3).

In comparison to the original CoVarScan validation study, the high level of performance achieved by the assay was upheld when evaluated using Omicron specimens. Positive percent agreement in the original study was 95.0% (3,210/3,378) compared with 96.5% (4,744/4,918) for the current Omicron study. The original study used 3544 clinical respiratory specimens, the vast majority of which were of the Delta lineage ($n = 2,820$, 79.6%). The original study also examined a small number of specimens of the Omicron lineage ($n = 309$, 8.7%), which were reused in the current analysis. The rest of the specimens used in the original validation were of the Alpha, Beta, Gamma, Lambda, Mu, and Iota linages (13).

**TABLE 1** Concordance counts between CoVarScan and WGS for each variant are underlined[a,d,e]

| CoVarScan PCR | SARS-CoV-2 whole-genome sequence (reference) | | | | | | | | | |
|---|---|---|---|---|---|---|---|---|---|---|
| | Delta | BA.1 | BA.2 | BA.2.12.1 | BA.2.75 | BA.4/5 | XBB | HV.1 | BA.2.86 | Below LOD[b] |
| Delta | 4 | 2 | | | | | | | | |
| BA.1 | | 279 | 14 | | | | | | | |
| BA.2 | | 1 | 276 | 10 | | | | | | |
| BA.2.12.1 | | | 2 | 366 | | 2 | | | | |
| BA.2.75 | | | 1 | | 42 | 1 | 1 | | | 0 |
| BA.4/5 | | 2 | 13 | 2 | 1 | 1,504 | 3 | | | 54 |
| XBB | | | | | | 4 | 1,634 | 108 | | 2 |
| HV.1 | | | | | | 7 | | 134 | | 1 |
| BA.2.86 | | | | | | | | | 505 | 24 |
| Other variant[c] | | 2 | 6 | 11 | 11 | 26 | 12 | 0 | 0 | 0 |

[a]Discordant results are shown in other cells when applicable.
[b]Below LOD: WGS is less sensitive than CoVarScan so when its results are not available as a comparator method, the samples are excluded from analysis ($n = 81$, 1.6% of WGS samples failed).
[c]Other variant: classification given when an additional, unexpected mutation was present, or to designate an instance of co-infection of more than one variant. Excluded from analysis.
[d]Positive percent agreement = (concordant/ total) = 4,744/4,918 = 96.5%.
[e]Gray area indicates identify of Omicron sub-variant.

**TABLE 2** Mutational signature: eight targets used by CoVarScan: five recurrently deleted regions (S:RDR1, S:RDR2, S:RDR3-4, ORF1A, ORF8) and three SNPs (S:N501(MUT), S:L452(MUT), S:E484(MUT))[a]

| Variant | S:RDR1 | S:RDR2 | S:RDR 3-4 | ORF1A | ORF8 | S:N501 (MUT) | S:L452 (MUT) | S:E484 (MUT) |
|---------|--------|--------|-----------|-------|------|--------------|--------------|--------------|
| Omicron (BA.1) | Δ6 bp | Δ9 bp | ins6 bp | Δ9 bp | Δ9 bp | MUT | | |
| BA.2 | | | | Δ9 bp | Δ9 bp | MUT | | |
| BA.2.12.1 | | | | Δ9 bp | Δ9 bp | MUT | MUT | |
| BA.4/BA.5 | Δ6 bp | | | Δ9 bp | Δ9 bp | MUT | MUT | |
| BA.2.75 | | | | Δ9 bp | Δ9 bp | MUT | MUT | |
| XBB | Δ3 bp | | (dropped) | Δ9 bp | Δ9 bp | MUT | | |
| BA.2.86 | Δ3 bp | Δ6 bp | Δ3 bp | Δ9 bp | Δ9 bp | MUT | (dropped) | (dropped) |

[a]Omicron (BA.1) mutational signature: ORF1A Δ9 bp, RDR2 Δ9 bp, RDR1 Δ6 bp, RDR3-4ins6 bp, and S:N501(MUT). BA.2 mutational signature: ORF1A Δ9 bp, ORF8 Δ9 bp, S:N501(MUT). BA.2.12.1 mutational signature: ORF1A Δ9 bp, ORF8 Δ9 bp, S:N501(MUT), S:L452(MUT). BA.4/BA.5 mutational signature: ORF1A Δ9 bp, ORF8 Δ9 bp, RDR1 Δ6 bp, S:N501(MUT), S:L452(MUT). BA.2.75 mutational signature: ORF1A Δ9 bp, ORF8 Δ9 bp, S:N501(MUT), S:L452(MUT). XBB mutational signature: ORF1A Δ9 bp, ORF8 Δ9 bp, S:RDR1 Δ3 bp, S:RDR3-4 dropped (<1:4 height of E484), S:N501(MUT). BA.2.86 mutational signature: ORF1A Δ9 bp, ORF8 Δ9 bp, RDR1 Δ3 bp, RDR2 Δ6 bp, RDR3-4 Δ3 bp, S:N501(MUT, Δ3 bp), S:E484(MUT, Δ3 bp), S:L452(MUT).

### Accuracy of nares vs nasopharyngeal samples

Furthermore, we tested 1,316 anterior nares samples and 4,691 nasopharyngeal samples over a select portion of the validation time period (8 February 2022 to 28 January 2023, Table 4). Of these anterior nares samples, 113 failed CoVarScan (8.5%) and 714 failed WGS (54%). Of the nasopharyngeal samples, 112 failed CoVarScan (2.4%) and 2,650 failed WGS (56%). In total, 489 nasal swabs and 1,919 nasopharyngeal specimens had both passing WGS and CoVarScan results for comparison, which showed 96.9% and 98.0% concordance, respectively. This demonstrates a high level of accuracy in a large number of samples including newly emergent Omicron subvariants.

### Adaptability to new, highly mutated variants

We were able to detect and track the highly mutated BA.2.86 variant (16). The number of spike gene differences from XBB was as large a divergence as Omicron from Delta. Similarly, a new variant named BA.2.87 was also recently reported in South Africa (17), and it is also highly mutated but contains a unique mutational signature identifiable by CoVarScan.

### CoVarScan has improved sensitivity compared with sequencing

Due to sample quality or low RNA amount, not all specimens were sent for WGS. We did not send specimens for WGS if CoVarScan amplified ≤3 targets or had CT values > 30 as most of these specimens fail WGS due to poor coverage (<90% coverage at 10×) or poor-quality metrics (e.g., higher personal mutations, increased frameshift variants).

Within our cohort, not all tests provided a CT value. Of the samples that returned with a CT value ($n = 2,311$), 379 specimens had CT values > 30, with an average CT value of 33.0. Within these lower RNA concentration samples, CoVarScan was able to assign a result for 353 specimens (353/379, 93.1%, for all specimens CT > 30). In this same range of CT values, only 87 cases yielded a WGS result (87/379, 23%, for all specimens CT > 30). Therefore, in a real-world setting, CoVarScan provides approximately 12% improved sensitivity compared to WGS (266/2311). To further estimate this improved sensitivity (because several instruments do not report CT values), we counted CoVarScan results without CT values that amplified the minimum number of targets required for interpretation (three). These samples were not sent for WGS ($n = 1,069$) as it was likely to fail. Thus, CoVarScan produces 17.8% (1,069/6,009) more test results than WGS would by this measure. Overall, we estimate CoVarScan is 22.2% (1,069 + 266/6,009) more sensitive than WGS methods.

**TABLE 3** Agreement and Disagreement rates between CoVarScan and WGS

| Percent positive agreement (concordant results/ total results) | Discrepancy rate (discrepant results/total results) |
|---|---|
| 96.5% (4,744/4,918) | 3.5% (174/4,918) |

**TABLE 4** Concordance for samples taken from the anterior nares was 96.9% while concordance for samples taken from the nasopharynx was 98.0%

| Sample type | Samples tested | Failed CoVarScan | Failed WGS | CoVarScan + WGS reportable | Concordant reportable result | % Concordant |
|---|---|---|---|---|---|---|
| Nares anterior | 1,316 | 113 | 714[a] | 489 | 474 | 96.9 |
| Nasopharyngeal | 4,691 | 122 | 2,650[a] | 1,919 | 1,881 | 98.0 |

[a]Timeframe: 8 February 2022 to 28 January 2023.

## DISCUSSION

The Omicron variant was discovered in November 2021 as a highly mutated variant with characteristic S-gene targeting failure upon routine PCR testing (18). Over the ensuing years, multiple subvariants have arisen. Tracking these variants has been performed by WGS in the United States, but the United Kingdom has moved toward genotyping most cases with occasional WGS. Given that most impactful variants are significantly different from circulating strains, genotyping approaches like CoVarScan could be epidemiologically useful. Furthermore, when variants have a clinically significant impact on mAb therapy decisions, results must be immediately available. These decisions are not feasible with WGS but are with CoVarScan, which can be performed in 4 h.

Overall concordance between CoVarScan and WGS was 96.5%. One reason for discordant results centered around closely related BA.2/BA.2.12/BA.2.12.1 variants. This could have arisen from carryover contamination: compared with CoVarScan, WGS involves an increased number of handling steps, an extra round of PCR, and higher potential for carryover or aerosolization from very high-titer specimens. In addition to limiting the frequency of WGS runs to once a week while CoVarScan runs may occur multiple times a week, the nature of how WGS is performed gives rise to technical issues. Positive and negative quality controls were included on each run of WGS and CoVarScan, but their performances differed significantly for each platform. Issues were more common with WGS. False-positive contamination of negative controls occurred at a high rate, 30% of WGS runs (16/54). False-negative results in positive control specimens were found in 5% (3/54) WGS runs. In contrast, false-positive and -negative results were only observed in 1.7% (3/174) and 0.5% (1/174) of CoVarScan runs, respectively. Another contributor to discordance between results was the dynamic nature of the naming convention for SARS-CoV-2 lineages, known as Phylogenetic Assignment of Named Global Outbreak Lineages (Pango) nomenclature, as there is lag time between calling of a new variant by Pango and updating results with new associated nomenclature (19).

Separately, some post-analytic issues were encountered with bioinformatic analyses performed on sequencing data. To call a mutation, the pipeline assumed that the variant allele frequency should be >50% (as most viral infections are clonal). However, we found 58 instances of simultaneous detection of two unique Omicron variants by CoVarScan (co-infection). WGS is not set up to call such co-infections, and a variant could be inaccurately assigned due to this co-infection scenario. However, upon manual inspection of sequencing files in Integrative Genomics Viewer (https://igv.org/), evidence of CoVarScan's detection of co-infection was confirmed each time. Co-infections were found most often when multiple variants were circulating at the same time. Furthermore, as described above, incomplete sequencing coverage prevented accurate lineage assignment in cases where few variants made the difference (e.g., BA.2.12.1 vs BA.2). In contrast, the eight-site combination of mutations of CoVarScan is simple, easier to interpret, and obvious when a site is not covered, unlike WGS. Therefore, WGS suffers from several post-analytic challenges introduced by bioinformatic pipelines and ever-evolving Pangolin lineages.

CoVarScan's strengths are highlighted compared with other PCR-based tests, which mostly focus on simple detection of SARS-CoV-2 and may use primers that target a variety of viral genes. While RT PCR-based assays are easily implemented due to the lack of need for specialized equipment/reagents, many commercial RT-PCR kits are unable to effectively and efficiently differentiate between emerging variants of SARS-CoV-2 (9). CoVarScan has been validated using a large sample size and is relatively simple,

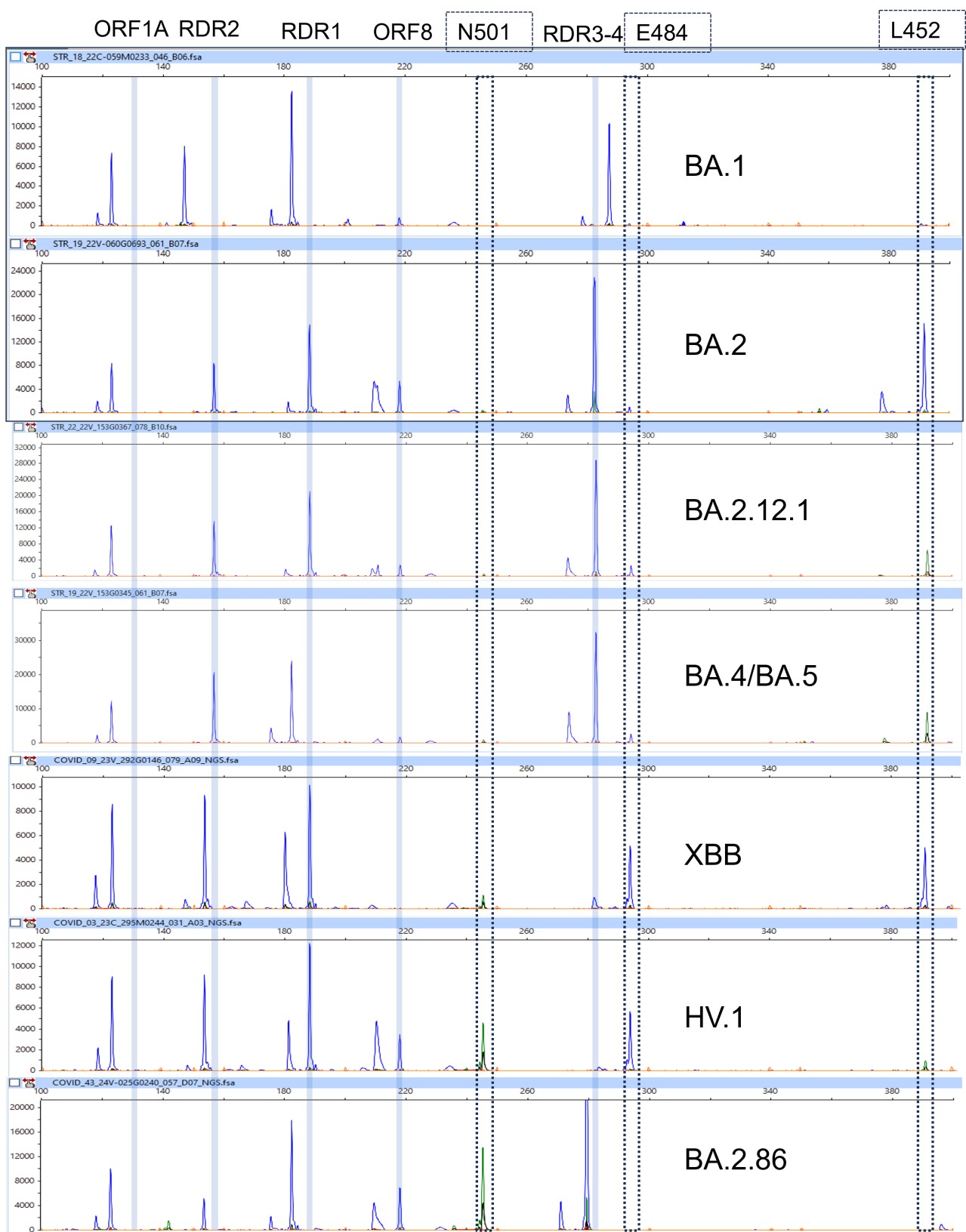

**FIG 1** Electrophoretogram of detected Omicron variants. Blue bars indicate reference gates compared with the original Wuhan strain. Dotted boxes denote SNP variants of the spike gene: N501, E484, and L452. The original Omicron variant (BA.1) characterized by ORF1A Δ9 bp, RDR1 Δ9 bp, RDR2 Δ6 bp, RDR3-4 ins6 bp, ORF8 Δ9 bp, E484$^{drop}$ and L452$^{drop}$. Electrophoretogram of BA.2 variant (distinguished from BA.1 by RDR1 WT, RDR2, WT, RDR3-4 WT, ORF8 Δ9 bp and

FIG 1 (Continued)

L452$^{WT}$), BA.2.12.1 (distinguished from BA.2 by L452$^{Mut}$), BA.4/BA.5 (distinguished from BA.2.12.1 by RDR2 Δ6 bp), XBB (distinguished from BA.4 by RDR1 Δ3 bp, RDR2$^{WT}$, RDR3-4$^{drop}$, E484$^{WT}$, and L452$^{WT}$), HV.1 (distinguished from XBB by L452$^{Mut}$), and BA.2.86 (distinguished from XBB by RDR2 Δ6 bp, RDR3-4 Δ3 bp, N501$^{Mut, high}$, L452$^{drop}$).

fast, and adaptable to emerging lineages, unlike other PCR-based approaches (4, 5, 8). CoVarScan did not require changing mutation-specific targets, unlike digital droplet PCR or microarray approaches (20–22). Lastly, CoVarScan mutational signatures can be predicted accurately from genomic variants *a priori*, unlike approaches using high-resolution melting analysis, which requires empiric testing (14, 23, 24).

A few limitations of the CoVarScan approach should be recognized. The assay was unable to distinguish between BA.4/BA.5 and BQ.1/BQ.1.1 (and related sublineages), themselves offshoots of the BA.5 lineage (25). Furthermore, CoVarScan uses only one sequencing chemistry. Lastly, the bioinformatics pipeline at our institution is custom and does not update as frequently as the database used by Pangolin.

In summary, we have verified our fragment analysis approach to SARS-CoV-2 VOC detection accommodates Omicron sublineages. CoVarScan can accurately distinguish different variants at a single-nucleotide resolution in a cost-effective, rapid, and scalable

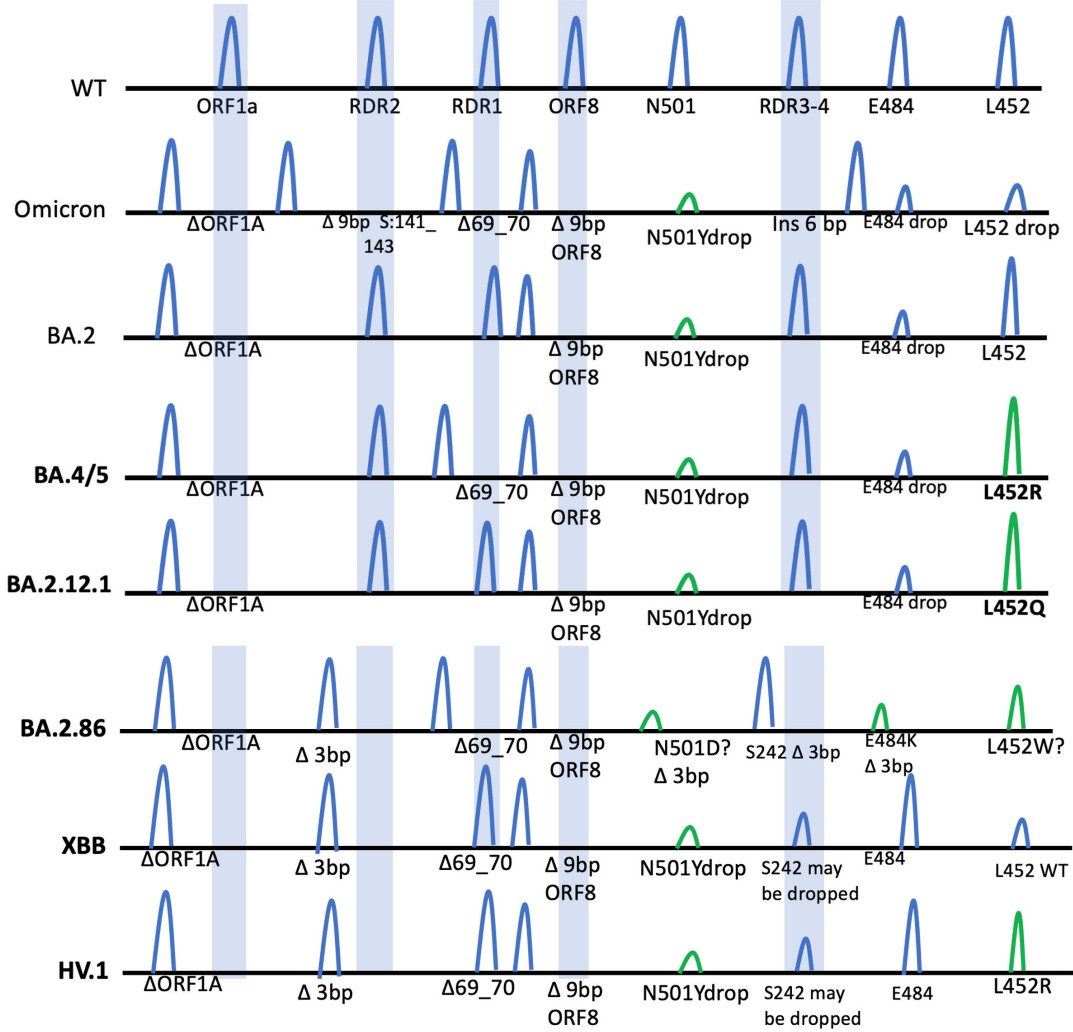

FIG 2   Diagram illustrating the location of each eight-plex target for each of the significant Omicron subvariants: BA.1, BA.2, BA.2.12.1, BA.4/BA.5, BA.2.75, XBB, HV.1, and BA.2.86.

manner. Moreover, CoVarScan does not require updating to detect new VOCs. Lastly, CoVarScan can perform as accurately as WGS with higher sensitivity, allowing its use as a tool to rapidly identify variants for epidemiological surveillance and clinical decision-making in the selection of effective therapeutic mAbs.

## ACKNOWLEDGMENTS

We thank the Texas Advanced Computing Center (https://www.tacc.utexas.edu) at The University of Texas at Austin for providing high-performance computing resources that have contributed to the research results reported within this study.

The Texas SARS-CoV-2 Variant Network Project is funded and supported by the Texas Department of State Health Services (DSHS) as part of a financial assistance award from the Centers for Disease Control and Prevention (CDC) of the U.S. Department of Health and Human Services (HHS) totaling $15 million, with 100% funded by CDC/HHS (HSC-SPH-21-05-0589: Texas COVID-19 Variant Network).

The contents are those of the author(s) and do not necessarily represent the official views of, nor an endorsement by, DSHS, CDC/HHS, or the US Government.

## AUTHOR AFFILIATIONS

[1]UT Southwestern Medical Center, Dallas, Texas, USA
[2]Methodist Health System, Dallas, Texas, USA

## AUTHOR ORCIDs

Kenneth Zhu ⓘ http://orcid.org/0000-0002-9335-2691
Jeffrey SoRelle ⓘ http://orcid.org/0000-0003-2588-6277

## FUNDING

| Funder | Grant(s) | Author(s) |
| --- | --- | --- |
| Texas Department of State Health Services (DSHS) | | Lenin Mahimainathan |
| | | Jeffrey SoRelle |

## AUTHOR CONTRIBUTIONS

Jeffrey SoRelle, Conceptualization, Data curation, Formal analysis, Funding acquisition, Investigation, Methodology, Project administration, Supervision, Writing – review and editing.

## DATA AVAILABILITY

A CSV file listing all samples analyzed in our study period (17 December 2021 to 31 January 2024) submitted to GISAID with collection date, GISAID ID, originating lab, and site of collection has been uploaded to the GitHub repository "utsw_gisaid_all_time" (https://github.com/knnzhu/utsw_gisaid_all_time). DOI: 10.5281/zenodo.14184055.

## ADDITIONAL FILES

The following material is available online.

Open Peer Review

**PEER REVIEW HISTORY (review-history.pdf).** An accounting of the reviewer comments and feedback.

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
