## [Reviewer comments · Microbiology Spectrum]

Microbiology Spectrum

Prospective Clinical Performance of CoVarScan in Identifying SARS-CoV-2 Omicron Subvariants

Kenneth Zhu, Manoj Sah, Lenin Mahimainathan, Chao Xing, Yan Liu, Karen Roush, Andrew Clark, and Jeffrey SoRelle

Corresponding Author(s): Jeffrey SoRelle, The University of Texas Southwestern Medical Center

Review Timeline:

Submission Date:	June 24, 2024
Editorial Decision:	August 22, 2024
Revision Received:	October 4, 2024
Accepted:	November 7, 2024

Editor: JJ Miranda

Reviewer(s): The reviewers have opted to remain anonymous.

Transaction Report:

DOI: <https://doi.org/10.1128/spectrum.01385-24>

Re: Spectrum01385-24 (Prospective Clinical Performance of CoVarScan in Identifying SARS-CoV-2 Omicron Subvariants)

Dear Dr. Jeffrey A SoRelle:

Thank you for the privilege of reviewing your work. Below you will find my comments, instructions from the Spectrum editorial office, and the reviewer comments.

Thank you for sharing your thorough validation study for a method to identify SARS-CoV-2 variants in a clinical setting. Spectrum recognizes the value of rigorous methodology, and the journal would be glad to receive a revised manuscript. Two reviewers have contributed thoughtful critiques toward that revision. Editorially, I would also like to add three notes:

- 1) Although I acknowledge that this involves a large number of sequences, it is general Spectrum policy that sequencing data be shared through a repository standard for the field.
- 2) Please clarify whether or not and if so how many samples examined in the 2022 Clinical Chemistry paper were also used in this study. The overlap between this and the previous publication occur at the peak of the Omicron wave. The statement "Specimens utilized are largely the same as those described in our previous study detailing the validation of CoVarScan [13]" requires more information.
- 3) It would benefit a reader if you presented a short discussion explicitly comparing the statistics and results of this study with those of the 2022 Clinical Chemistry paper.

Revision Guidelines

Sincerely,

JJ Miranda
Editor
Microbiology Spectrum

Reviewer #1 (Comments for the Author):

In the paper titled "Prospective Clinical Performance of CoVarScan in Identifying SARS-CoV-2 Omicron Subvariants", the authors assess CoVarScan, a multiplex fragment analysis technique designed for the rapid and accurate identification of SARS-CoV-2 Omicron variants. The study demonstrates CoVarScan's 98.4% concordance with WGS and its capability to identify various Omicron subvariants, including the newly emergent BA.2.87. The manuscript presents CoVarScan as a scalable and cost-effective alternative to WGS, offering significant benefits for epidemiological surveillance and clinical decision-making regarding therapeutic monoclonal antibodies. Overall, the findings highlight CoVarScan's potential for efficient pandemic monitoring and variant identification.

The manuscript is well-written, and the number of samples analyzed justifies the results. However, this reviewer does not see the novelty of this work. The same group published a paper in 2022, which is also cited in the current manuscript, where they concluded that multiplex fragment analysis is adaptable, rapid, and has similar accuracy to WGS for classifying SARS-CoV-2 variants (10.1093/clinchem/hvac081). Omicron was one of the VOCs that they were able to differentiate.

Reviewer #2 (Public repository details (Required)):

4918 samples were sequenced during the study. If possible, sequences should be made available in a public repository, and a dataset with a doi created.

Reviewer #2 (Comments for the Author):

Comments in attached file

This study evaluate the accuracy of the CoVarScan assay, and compare its performance and advantages against the standard Whole Genome Sequencing (WGS).

The study include a large number of samples collected over a long period, and covering several variants, including related sub-lineages. The discrepancies in the results obtained with the 2 different approaches are discussed and explained. Most of the discrepancies concern related sub-lineages, especially BA.2.

The results support the author main points when comparing against WGS. However, the tables presenting the results could be improved.

I only have a few remarks and suggestion regarding this manuscript:

- Most of my remarks are regarding Table1
 - Table 1 is not called in the main text/results.
 - It was not clear at first look that Table 1 show the results after resolving discrepancies
 - Numbers in Table 1 do not matches:
 - $4,744/4,914=96.54\%$, not 98.6%
 - I counted 4,744 concordant results, for 4,918 total (96.46%). 4,918 is the total number of samples sequenced according to the main text. I am not sure where/why we lost 4 samples in the last line of Table 1 legend.
 - Percentages given in the text need to match percentages in the table legend
- The authors should discuss the issue regarding BA.4/5 vs. HV.1, especially the 108 HV.1 counted as BA.4/5
- In the Results section, paragraph "*Accuracy of CoVarScan*", please indicate how many samples fall in each category (concordant vs. discrepant) to match the given percentages. Same remark regarding Table 3
- In the Results section, first paragraph, the sentence "To investigate..." need to be corrected
- The last 2 paragraphs (Technical issue and Analytic issue) would fit better in the discussion
- If possible, sequences from WGS should be made available on a public repository. For an easy access from the manuscript, a dataset with a doi/link could be created.

We thank the reviewers for their time and considerate and thoughtful responses for this report. We are pleased to address their comments and questions to improve the quality of this manuscript.

Editor comments:

- 1) Although I acknowledge that this involves a large number of sequences, it is general Spectrum policy that sequencing data be shared through a repository standard for the field.
Data made available through upload to GISAID of 3239 samples (identifiers attached as csv document). This represents the vast majority of the dataset, which could be uploaded by the sponsor.
- 2) Please clarify whether or not and if so how many samples examined in the 2022 Clinical Chemistry paper were also used in this study. The overlap between this and the previous publication occur at the peak of the Omicron wave. The statement "Specimens utilized are largely the same as those described in our previous study detailing the validation of CoVarScan [13]" requires more information.
Clarification added within sentence referenced above and in paragraph added under "Accuracy of CoVarScan" section
- 3) It would benefit a reader if you presented a short discussion explicitly comparing the statistics and results of this study with those of the 2022 Clinical Chemistry paper.
Paragraph added under "Accuracy of CoVarScan" section

R1

This study evaluate the accuracy of the CoVarScan assay, and compare its performance and advantages against the standard Whole Genome Sequencing (WGS). The study include a large number of samples collected over a long period, and covering several variants, including related sub-lineages. The discrepancies in the results obtained with the 2 different approaches are discussed and explained. Most of the discrepancies concern related sub-lineages, especially BA.2. The results support the author main points when comparing against WGS. However, the tables presenting the results could be improved.

I only have a few remarks and suggestion regarding this manuscript:

- Most of my remarks are regarding Table1
 - Table 1 is not called in the main text/results.
Added sentence about table 1 in beginning of "Accuracy of CoVarScan" section
"Table 1 organizes concordant (in green) and discordant results by subvariants called by both CoVarScan and WGS after resolving for discrepancies due to technical and bioinformatic reasons."
 - It was not clear at first look that Table 1 show the results after resolving discrepancies
Made explicit in sentence mentioned above
 - Numbers in Table 1 do not matches:

- 4,744/4,914=96.54%, not 98.6%

Corrected

- I counted 4,744 concordant results, for 4,918 total (96.46%). 4,918 is the total number of samples sequenced according to the main text. I am not sure where/why we lost 4 samples in the last line of Table 1 legend.

Corrected in table legend and main text – 4,918 samples were analyzed with a concordance of 96.5% (3.5% discordant)

- Percentages given in the text need to match percentages in the table legend

Corrected

- The authors should discuss the issue regarding BA.4/5 vs. HV.1, especially the 108 HV.1 counted as BA.4/5

This is a misplaced number, and the 108 discrepant results should be between HV.1 and XBB.

The principal difference between the XBB and HV.1 variants was a SNP variant in the spike protein: L452R. This primer location is the longest amplicon at ~400 base pairs long, so it can drop out if RNA is degraded or with low viral RNA. When this happens, the XBB and HV.1 variants are indistinguishable by CoVarScan, but the sequencing approach can use other mutations to distinguish the variants.

- In the Results section, paragraph "Accuracy of CoVarScan", please indicate how many samples fall in each category (concordant vs. discrepant) to match the given percentages. Same remark regarding Table 3

Numbers added

- In the Results section, first paragraph, the sentence "To investigate..." need to be corrected

Fixed and edited for clarity

- The last 2 paragraphs (Technical issue and Analytic issue) would fit better in the discussion

Technical issues paragraph merged with paragraph starting with "Overall concordance..." in Discussion section to resolve redundancy

Analytical issues paragraph moved to under paragraph mentioned above

- If possible, sequences from WGS should be made available on a public repository. For an easy access from the manuscript, a dataset with a doi/link could be created. Our site does not have the ability to upload files to GISAID, but they were uploaded by our COVID-19 Network Coordinating center. There are 3,239 cases submitted to GISAID from February 2022. We will submit a CSV file indicating the collection date and GISAID submission ID. Lab identifiers and other potential PHI have been redacted. Sorry that a link could not be created but the samples have been uploaded progressively in multiple batches along with data from multiple sites, which is not under our control.

Re: Spectrum01385-24R1 (Prospective Clinical Performance of CoVarScan in Identifying SARS-CoV-2 Omicron Subvariants)

Dear Dr. Jeffrey A SoRelle:

Thank you for your detailed study further validating a practical PCR-based alternative to whole-genome sequencing for the rapid identification of contemporary SARS-CoV-2 variants.

Your manuscript has been accepted, and I am forwarding it to the ASM production staff for publication. Your paper will first be checked to make sure all elements meet the technical requirements. ASM staff will contact you if anything needs to be revised before copyediting and production can begin. Otherwise, you will be notified when your proofs are ready to be viewed.

Sincerely,
JJ Miranda
Editor
Microbiology Spectrum